# Management of Type 2 Diabetes Mellitus in Elderly Patients with Frailty and/or Sarcopenia

**DOI:** 10.3390/ijerph19148677

**Published:** 2022-07-16

**Authors:** Jaime Sanz-Cánovas, Almudena López-Sampalo, Lidia Cobos-Palacios, Michele Ricci, Halbert Hernández-Negrín, Juan José Mancebo-Sevilla, Elena Álvarez-Recio, María Dolores López-Carmona, Luis Miguel Pérez-Belmonte, Ricardo Gómez-Huelgas, Maria Rosa Bernal-López

**Affiliations:** 1Unidad de Gestión Clínica de Medicina Interna, Hospital Regional Universitario de Málaga, Universidad de Málaga (UMA), 29010 Málaga, Spain; 0610880681@uma.es (J.S.-C.); almudena.lopez.sampalo.sspa@juntadeandalucia.es (A.L.-S.); lidia.cobos.sspa@juntadeandalucia.es (L.C.-P.); michele.ricci.sspa@juntadeandalucia.es (M.R.); halberthn@uma.es (H.H.-N.); juanj.mancebo.sspa@juntadeandalucia.es (J.J.M.-S.); elenaalvarez@correo.ugr.es (E.Á.-R.); mariad.lopezcarmona.sspa@juntadeandalucia.es (M.D.L.-C.); lmiguel.perez.sspa@juntadeandalucia.es (L.M.P.-B.); 2Instituto de Investigación Biomédica de Málaga (IBIMA-Plataforma Bionand), 29590 Málaga, Spain; 3Centro de Investigación Biomédica en Red Fisiopatología de la Obesidad y Nutrición (CIBERobn), Instituto de Salud Carlos III, 28029 Madrid, Spain

**Keywords:** type 2 diabetes mellitus, elderly, sarcopenia, frailty

## Abstract

The life expectancy of the population is increasing worldwide due to improvements in the prevention, diagnosis, and treatment of diseases. This favors a higher prevalence of type 2 diabetes mellitus (T2DM) in the elderly. Sarcopenia and frailty are also frequently present in aging. These three entities share common mechanisms such as insulin resistance, chronic inflammation, and mitochondrial dysfunction. The coexistence of these situations worsens the prognosis of elderly patients. In this paper, we review the main measures for the prevention and management of sarcopenia and/or frailty in elderly patients with T2DM.

## 1. Introduction

The world population’s average age is increasing. The elderly population has a higher burden of diseases and comorbidities compared to the younger population as well as an accentuated heterogeneity among the pathologies they suffer from [1]. One of these entities is T2DM and its micro and macrovascular complications. Approximately half a billion people are living with diabetes worldwide, which means that over 10.5% of the world’s adult population now have T2DM. Currently, around half of patients with T2DM are over 65 years of age [1,2]. The prevalence of T2DM has also increased in the aging population in the last 50 years, and it is expected to continue increasing in the coming decades [2].

We currently have a large arsenal of treatments for the management of T2DM. Therapeutic guidelines vary depending on each patient since glycemic control is recommended to be less strict in this group of older patients. Furthermore, the multiple comorbidities, such as sarcopenia and frailty, that these patients usually present influence the therapeutic scheme of the elderly compared to the younger population [3]. A global and comprehensive evaluation of the elderly patient with T2DM is essential, with the aim to achieve a good quality of life, preserve the functionality of the patient, and avoid complications, mainly hypoglycemia [4].

On the other hand, frailty is a clinical syndrome associated with aging, present in a large number of elderly patients. Frailty, which is frequent in diabetic patients, consists of a decrease in homeostatic reserves and is responsible for greater vulnerability to stressors. Thus, it should be considered a condition of great interest for public health [5].

Another complication that occurs in aging patients is sarcopenia, defined as a geriatric syndrome characterized by decreased skeletal muscle mass and function. Sarcopenia is emerging as a major health care problem in T2DM patients, and antidiabetic agents may have some impact on muscle mass and performance in T2DM [6].

Currently, there is no consensus on defining an individual as elderly. In this document, as in most published documents, being over 75 years of age or presenting frailty are used as an operational definition [7].

Multiple advances have been published in the management of T2DM. This document focuses on the review of the therapeutic aspects of T2DM in elderly patients with frailty and/or sarcopenia.

## 2. Sarcopenia and Frailty in Older T2DM Patients

The presence of T2DM in the elderly is becoming more frequent. These patients usually present multiple comorbidities, some of them are related to the natural evolution of diabetes (neuropathy, nephropathy, retinopathy, etc.) and other entities are independent of the evolution of diabetes, although they may be related. Frailty is one of them, a standardized phenotype in older adults with a predictive validity for adverse outcomes in geriatric patients. It is an emerging global health burden with important implications for clinical practice and public health [8]. 

Diabetes mellitus is characterized by a state of chronic hyperglycemia associated with alterations in carbohydrate, fat, and protein metabolism leading to a deficit of insulin secretion and to insulin resistance, which are involved in muscle protein loss. Elevated levels of cytokines, such as TNFα, IL-1, IL-5 or IL-6, common in patients with T2DM, can induce insulin resistance. In addition, the usual mitochondrial dysfunction in diabetic patients favors altered lipid oxidation, elevated lipid in muscle cells, and insulin resistance, which increases the development of frailty and sarcopenia [9]. Moreover, loss of muscle mass secondary to sarcopenia and age also leads to metabolic dysregulation resulting in reduced insulin sensitivity, altered oxidative defenses, and decreased mitochondrial function. Age-related alterations in the hypothalamus–pituitary–testicular, hypothalamus–pituitary–adrenal, and insulin-like growth factor I (IGF-1) and type 1 IGF receptor (IGFR1r) axes increase the loss of muscle strength [10]. Testosterone and IGF-1, hormones involved in muscle protein synthesis, are decreased in patients with T2DM. In addition, frequent alterations in the adipocyte hormone leptin and ghrelin in older age also interfere negatively with plasma insulin levels [11]. All these pathophysiological changes explain how diabetes mellitus, advanced age, frailty, and sarcopenia are interrelated (Figure 1).

In order to diagnose a patient with frailty, Fried’s frailty criteria are widely used [7]. According to these criteria, the diagnosis of frailty is established if the patient meets three of the following items: unintentional weight loss, exhaustion, muscle weakness, motor slowness, and low activity (Table 1). In addition, T2DM is a risk factor for the development of frailty. Therefore, nutritional education and physical activity to control glycemic levels are effective in maintaining functional autonomy [12].

Frailty is determined by genetic, epigenetic, and environmental factors, with insulin resistance, arteriosclerosis, brain white matter lesions, chronic inflammation, oxidative stress, or mitochondrial dysfunction being some of most common mechanisms of frailty in older adults with T2DM [13]. Due to the high prevalence of T2DM in frail older patients, and considering the complexity of T2DM and the burden of associated comorbidities, it is important to identify this group of vulnerable patients who require close follow-up with the aim to implement therapeutic measures and intervention strategies to avoid a further deterioration in functional status [14].

In addition, frailty is associated with mortality and hospitalization in patients with diabetes. Accidental falls are one of the main factors linking frailty with hospitalization. Fragile patients have an increased risk of falls [15]. Falls are also common in diabetic patients, and situations as hypoglycemia contribute even more to this relationship [16]. It is well known that both frail and diabetic patients have more complications and worse prognosis during hospital admissions. Therefore, it is greatly important to prevent this type of complication in frail and diabetic patients [17].

Increasing attention is being paid to sarcopenia due to its strong impact on the quality of life of elderly patients with T2DM [18]. Sarcopenia is defined as a muscle disease (muscle failure) in relation to adverse muscle changes across the lifetime, being common in elderly and focusing on low muscle strength as the main characteristic. The most common causes of sarcopenia are listed in Table 2. There are different ways to detect a low muscle quantity and quality to confirm the sarcopenia diagnosis [19]. The three main conditions in the differential diagnosis of sarcopenia are malnutrition, cachexia, and frailty, although they often coexist [20].

Sarcopenia has a higher prevalence in patients with T2DM (prevalence ranging from 5 to 50%) than in patients without T2DM [21]. Sarcopenia associated with frailty and the microvascular and macrovascular complications of diabetes are emerging as an important category of complications leading to disability, dependency, and increased mortality [18]. All this has a high impact on the quality of life of patients, affecting their physical and psychosocial health and, consequently, becoming an important public health problem [22]. Quality of life is the main objective in older patients; therefore, early detection of frailty and sarcopenia are key aspects in the management of older patients in general and diabetics in particular [23].

T2DM, sarcopenia, and frailty generate chronic systemic proinflammatory states that produce immunological changes in the immune systems of patients. Immunosenescence is associated with a worse clinical course. It favors the acquisition of infectious diseases and poor response to vaccination, among others [24]. On the other hand, the proinflammatory state favors the development of tissue damage, sarcopenia, and frailty. All these situations intertwined with each other hamper the management of these patients [25,26].

Another complication commonly observed in patients with T2DM and sarcopenia is sarcopenic obesity, defined by the concomitant presence of sarcopenia and obesity. The prevalence of sarcopenic obesity is higher in diabetic patients. Physical exercise and a healthy diet are the main treatment for sarcopenic obesity, accompanied by antidiabetic drugs individually prescribed by clinicians for the treatment of T2DM [27].

## 3. Glycemic Control Goals in the Elderly Patient

Therapeutic guidelines on the management of diabetes mellitus usually recommend strict glycemic control in young patients to avoid long-term diabetic complications. For a few years, much has been said about glycemic control in elderly patients. The risks of intense and strict treatment may outweigh the benefits of good glycemic control in this patient profile. Because of that, glycemic control goals in elderly patients should be adapted to their functional status, cognitive status, comorbidities, and life expectancy [4].

Not only hypoglycemia or hyperglycemia can cause complications. Glycemic variability is also important due to the complications derived from it in older patients Therefore, it is important to maintain as little glycemic variability as possible in these patients [28].

It is recommended to maintain a more conservative therapeutic approach in these elderly patients with sarcopenia and/or frailty since they do not benefit from strict glycemic control. This glycemic control should always be based on the patient’s functional status [29]. In these patients, who are usually affected by polypharmacy, treatment should be individualized and simplified to achieve better adherence. It is essential to avoid hypoglycemia due to its consequences. Elderly patients often have asymptomatic hypoglycemia due to the absence of adrenergic alarm symptoms. On other occasions, patients present cognitive impairment, which makes clinical suspicion difficult. It cannot be forgotten to adjust the dose of hypoglycemic treatments to the renal function of the patient. Hypoglycemia in elderly patients favors the development of cognitive impairment [30], confusional syndrome, stroke, falls and fractures, and higher mortality [31]. In fact, hypoglycemia secondary to the use of antidiabetics is an important cause of admission to the emergency room due to pharmacological adverse effects [32,33]. Therefore, it is important to avoid drugs that induce hypoglycemia, in addition to other symptoms such as hyporexia or digestive symptoms, in this patient profile.

In summary, treatment planning for T2DM in elderly subjects should be based on a comprehensive geriatric assessment. The priority objective is avoiding symptomatic hypoglycemia and hyperglycemia; maintaining an HbA1c of 7.5–8.5% and fasting blood glucose levels between 90–100 mg/dL and 150–180 mg/dL are reasonable goals in frail elderly patients with T2DM [4].

## 4. Non-Pharmacological Management

The key to improving outcomes for patients with sarcopenia and frailty is early recognition and intervention. The evaluation and detection of the deterioration of their physical function and activities of daily life in older adult patients must be actively performed by health personnel. The main management of sarcopenia is physical exercise in combination with nutritional intervention [34].

### 4.1. Physical Activity

Both aerobic and resistance exercise prevent and treat the decline in muscle mass and strength that occur with age [35]. These exercises must be individualized in the elderly. It is known that resistance training positively influences the neuromuscular system and increases hormone concentrations and the rate of protein synthesis [36]. In addition to improving functional independence, self-esteem, and quality of life, variable-intensity exercise programs have been shown to be effective in the elderly with diabetes by improving glycemic control [37]. Resistance training, even in the absence of aerobic exercise, improves the glycemic profile and reduces cardiovascular risk. On the other hand, low strength and muscle quality is associated with increased adiposity and chronic hyperglycemia [38]. It is important to remember that glucose levels should be monitored before and after the exercise session to reduce the fear of exercise-induced hypoglycemia [39].

Multiple types of physical exercise have been studied. A warm-up of approximately 5 min and a 3 min cool-down is recommended. Depending on the patient’s baseline physical condition, exercises such as walking at low intensity or, if not possible, riding a stationary bicycle may be performed. Other exercises may include wide leg squats, standing leg curls, hip extension, or hip flexion. For patients with severe physical impairment, simpler exercises such as knee extension, ankle circles, arm raise, chair push (triceps extension), tennis ball squeeze, seated neck turn, or leg circles may be performed [40].

### 4.2. Nutrition

Nutrition is another fundamental pillar in the management of these patients. Intake should be individualized based on nutritional status, physical activity level, disease status, and tolerance [13], but in general, the European Society for Clinical Nutrition and Metabolism guidelines on clinical nutrition and hydration in geriatrics recommend an energy intake of about 30 kcal/kg body weight/day in aging [41].

A protein intake of 1.0–1.2 g/kg body weight per day is recommended to maintain and restore body mass and function in patients older than 65 years [42]. Although an adequate diet to maintain good glycemic control is important in subjects with T2DM, malnutrition should be avoided in older patients, especially in patients with frailty or sarcopenia [15].

Vitamin D helps improve muscular mass and strength. Serum vitamin D deficiency is associated with frailty and sarcopenia [43]. Muscle mass and function of the lower extremities are improved as a result of administration of vitamin D in conjunction with protein or physical exercise in elderly patients with sarcopenia [44]. Currently, we do not have studies aimed at diabetic patients at high risk of sarcopenia, but a sufficient intake of vitamin D accompanied by protein and exercise could be recommended in elderly patients with diabetes, especially those with sarcopenia [13].

There is limited evidence with other vitamins such as C, E, B_6_ and B_12_. Despite the high prevalence of vitamin B_12_ deficiency in asymptomatic older people, there is no formal recommendation for screening [45]. The relationship between the intake of these supplements and frailty has been studied in some observational studies [46]. There have been few clinical trials with supplements of these vitamins, without positive results [47,48]. Vitamin supplementation could have positive effects on cognitive function in patients with subclinical vitamin deficiency, but this should be investigated in future studies with elderly patients with diabetes as the evidence published is not clear.

Polyunsaturated fatty acids (PUFAs), especially omega-3 fatty acids (ω-3FA), including eicosapentaenoic acid (EPA) and docosahexaenoic acid, have been studied in older patients with diabetes. Observational studies suggest that these fatty acids may be important in preventing cardiovascular disease and sarcopenia in elderly patients [49]. On the other hand, clinical trials showed controversial results [50,51]. Therefore, further research in this field is necessary before interpreting these results.

Finally, the Mediterranean diet is widely considered as a healthy diet. It consists of vegetables, fish, and olive oil intake, among others. It is known to reduce the risk of cardiovascular events. Published meta-analyses recommend adherence to the Mediterranean diet to reduce the risk of frailty and functional disability, including patients with T2DM. Therefore, the Mediterranean diet is highly recommended in this population [52,53].

### 4.3. Other Recommendations

Although physical exercise and diet are two fundamental pillars in this population, other measures that provide benefits to the subjects are shown below. Multidisciplinary programs should be implemented to patients and family members or caregivers in order to prevent possible complications and train in the management of their baseline situation. Different methods of preventing falls and fractures must be shown to elderly diabetic patients with sarcopenia or frailty phenotypes [54].

Frail patients tend to have fewer social communications and poor networks, which are risk factors for depression [55]. Vascular depression is also associated with T2DM. Therefore, promoting a communications network is a priority to promote the best possible state of mental health [15,56].

Similarly, cognitive dysfunction is associated with physical frailty, diabetes, and sarcopenia. Insulin resistance is related to T2DM and implicated in the mechanisms of sarcopenia and frailty development, and it has been suggested to accelerate the pathology of Alzheimer’s disease. For this reason, cognitive stimulation programs and other measures that prevent and retard the development of dementia should be favored [57,58].

The key role of social ties, perceived support, and participation in social activities in promoting mental health in older age, especially among frail older adults, should be highlighted [59].

Finally, the prevention of hypoglycemia should always be a priority in frail diabetic patients because it is important to know how to recognize hypoglycemia and its management [15].

In conclusion, the implementation of multimodal and multidisciplinary interventions based on nutritional education and the promotion of physical activity should be performed with the aim of maintaining the greatest possible functional autonomy in T2DM patients with risk of frailty or sarcopenia, or some of them established [60]. It is important to mention that the benefit of reducing the risk of microvascular complications is lower than the likelihood of serious side effects due to hypoglycemia. However, patients with poorly controlled T2DM may suffer from acute complications of diabetes, such as dehydration, poor wound healing, or hyperglycemic hyperosmolar coma. Therefore, glycemic goals should primarily avoid these consequences [33]. Main non-pharmacological measures are summarized in Table 3.

## 5. Pharmacological Treatment

Multiple drugs have been approved for the treatment of diabetes mellitus. In contrast, little evidence is currently available for the pharmacological treatment of frailty. As described above, there are many interventions for the clinical management of frailty, such as physical exercise, a healthy diet, or complication prevention programs. The deprescription of unnecessary medications can also be beneficial. However, we do not have scientific evidence regarding the use of specific drugs against frailty [61]. Similarly, sarcopenia does not have specific pharmacological treatments, and its management is based on promoting physical activity and a healthy diet and avoiding associated complications [62]. However, some antidiabetic agents might have important roles in muscle physiology [6]. Below, the main groups of antidiabetic drugs and their involvement in sarcopenia and/or frailty in the elderly are reviewed.

### 5.1. Biguanides

Metformin is a drug widely used in the management of T2DM. Multiple studies support its use. Therefore, metformin is the first-line treatment in the vast majority of cases of T2DM. In elderly patients, metformin has not been studied in specific clinical trials, but it is commonly used due to the widely clinical experience [63]. Metformin does not usually cause hypoglycemia, and it is useful in patients with cardiovascular disease or stable heart failure. However, in the elderly, inconveniences such as digestive intolerance, dysgeusia, hyporexia, and vitamin b12 deficiency may appear [64]. In addition, its use must be adjusted to renal function.

Regarding sarcopenia, although the precise mechanisms are not clearly recognized, different studies show that metformin apparently has positive effects on both muscle mass and muscle strength [6,18]. For example, Aghili et al. suggested that newly diagnosed T2DM patients treated with metformin 1000 mg twice daily had a significant increase in skeletal mass index [65]. Other studies showed that metformin exposure was associated with a lower risk of frailty after adjustment for covariates [66]. However, clinical trials are necessary to obtain more consistent results [67].

### 5.2. Sulfonylureas

Sulfonylureas have been widely used drugs due to their low cost and the decrease in microvascular complications of T2DM [68]. They are ATP-sensitive potassium channel blockers. The release of insulin from the β cells of the pancreas is stimulated by these antidiabetic pills. They are currently disused drugs, especially in the elderly, due to the risk of hypoglycemia, the weight gain they generate, or the multiple pharmacological interactions they present (fibrates, allopurinol, salicylates, dicumarinics, methotrexate, beta-blockers, corticosteroids).

We have limited evidence in “in vivo” studies. In “in vitro” studies, it is observed that sulfonylureas favor muscle atrophy, thus they should be avoided in patients with sarcopenia or at risk of developing sarcopenia [69,70]. Based on animal experiments, the least effective sulfonylurea as an atrophic agent was glimepiride [70]. On the other hand, evaluation of frailty should be performed periodically in older patients taking sulfonylureas due to the risk of presenting complications because of the adverse effects of these antidiabetics [71,72].

### 5.3. Meglintinides

Glinides have a similar mechanism of action to sulfonylureas. Some differences compared to sulfonylureas are shorter circulating half-life, rapid absorption, elimination through the liver, and their action mainly on postprandial glucose levels. Like the sulfonylureas, these agents have a high risk of hypoglycemia, and therefore they are not recommended in the elderly [73,74]. Furthermore, these drugs must be used with meals, so their use is not recommended in frail patients with poor eating habits [74]. Within the glinides, few studies have been published on sarcopenia, although their muscular atrophic effect is known. Repaglinide is the most potent in vitro atrophic agent in animals [69], while other studies have shown that glibenclamide induces atrophy in animal experiments and in human patients [70].

### 5.4. Thiazolidinediones

Thiazolidinediones should not be drugs of first choice in elderly patients due to the comorbidities that these patients usually present. Some of their best-known adverse effects are decompensated heart failure and risk of fractures, frequent in this patient profile. This situation limits its use in older diabetic patients [75]. In contrast, thiazolidinediones do not produce lactic acidosis or hypoglycemia. Pioglitazone is still in use; however, rosiglitazone was suspended by the European Medicines Agency due to its side effects [76].

Being insulin sensitizers, some studies have shown beneficial effects of thiazolidinediones on muscle performance in diabetic patients. Pioglitazone improves energy in skeletal muscle by reducing intramyocellular lipid content and improving fatty acid metabolism [77]. In summary, these drugs are promising anti-atrophy agents, but the unfavorable benefit/risk profile associated with cardiovascular and adverse events limits their use in the elderly [78].

### 5.5. Incretins

Dipeptidyl peptidase-4 (DDP-4) inhibitors and glucagon-like peptide 1 (GLP-1) agonists are drugs recently incorporated into the therapeutic arsenal of T2DM.

DPP-4 inhibitors have few side effects and minimal risk of hypoglycemia [79]. They can be used safely, without risk of hypoglycemia, at any stage of chronic renal failure. All of them require dose adjustment in cases of moderate or severe renal insufficiency, except linagliptin, which undergoes biliary elimination. They do not modify body weight or present significant drug interactions [80]. They do not cause digestive intolerance either. These are all important advantages in elderly patients. DDP-4 inhibitors may increase muscle mass, although their mechanism is unclear. It could be related to ability to enhance GLP-1 action or inhibition of DPP-4 activity per se, or both [18].

GLP-1 agonists have demonstrated anti-oxidative and anti-inflammatory properties, along with anti-thrombotic effects, that could be helpful in frailty patients due to the high burden of cardiovascular comorbidities that these patients usually present [81,82]. However, although the mechanisms regarding the effects of GLP-1 agonists on skeletal muscle continue being a matter of debate, the weight loss and decreased appetite produced by GLP-1 agonists may have undesirable effects in the frail elderly, in whom hyporexia and malnutrition are common. Weight loss at the expense of lean mass can be counterproductive in patients with sarcopenia. They may also be associated with nausea, vomiting, and diarrhea [83]. Therefore, GLP-1 agonists could be used with caution in the elderly, but they should not be administered in the frail elderly.

### 5.6. Sodium-Glucose Cotransporter 2 (SGLT-2) Inhibitors

Its mechanism of action is to prevent glucose reabsorption in the renal tubule and induce glycosuria. This family of drugs has shown benefits in patients with heart failure, cardiovascular benefits in patients with established cardiovascular disease, and delayed progression of chronic kidney disease, usually frequent in elderly and diabetic patients [84,85,86]. SGLT2 inhibitors have proven to be a good therapeutic option for the treatment of DM2 in older patients. They have a low risk of hypoglycemia, reduce blood pressure and weight, and provide cardiovascular and renal safety/protection.

However, these medicines should be introduced with caution in the elderly who are taking other treatments such as insulin or hypotensive drugs [87]. It is important to avoid excessive volume depletion secondary to osmotic diuresis in adult patients [88]. In addition, due to its hypotensive effect, the existence of orthostatic hypotension, frequent in elderly diabetic patients, must be assessed [89]. Another complication that we must evaluate is the presence of genital mycoses or urinary infections that can complicate the evolution of frail patients.

There is currently no scientific evidence available on the effects of these drugs on muscle. Future studies are required to study whether SGLT2 inhibitors have atrophic/anti-atrophic effects [15].

### 5.7. Insulin

Insulin is the most powerful hypoglycemic drug known to date. Insulin-treated elderly diabetic patients are at increased risk of severe hypoglycemia, falls, and fractures. Therefore, the use of insulin in the elderly must be individualized. Oral treatments are preferable instead of insulin in older patients. Basal insulin analogs are preferable to human insulin because of their lower risk of hypoglycemia, mainly at night, although their cost is higher. If insulin must be used, it is preferable to use simple guidelines and avoid exhaustive glycemic controls [90,91].

Insulin stimulates muscle protein synthesis in young adults, but not in the elderly. Skeletal muscle atrophy is not preventable with insulin therapy, probably due to the condition of insulin resistance generated during aging [92,93]. There is no evidence of benefits of insulin treatment on sarcopenia in the elderly.

A summary of the main pharmacological recommendations in patients with frailty and T2DM is described in Table 4.

## 6. Conclusions

T2DM, sarcopenia, and frailty are mutually associated and often coexist. Their presence is a marker of poor prognosis in elderly patients. It is recommended to avoid excessive glycemic control, with the main objective of avoiding hypoglycemia and symptomatic hyperglycemias. Screening and early detection of sarcopenia and/or frailty are recommended using programs aimed to improve the prognosis and quality of life of patients. Probably, the lack of available data on new, ongoing studies that have not yet been published may be a limitation in our work. Further studies on the joint management of these three pathologies are recommended due to their increasing prevalence and the scarce evidence published to date.

## Figures and Tables

**Figure 1 ijerph-19-08677-f001:**
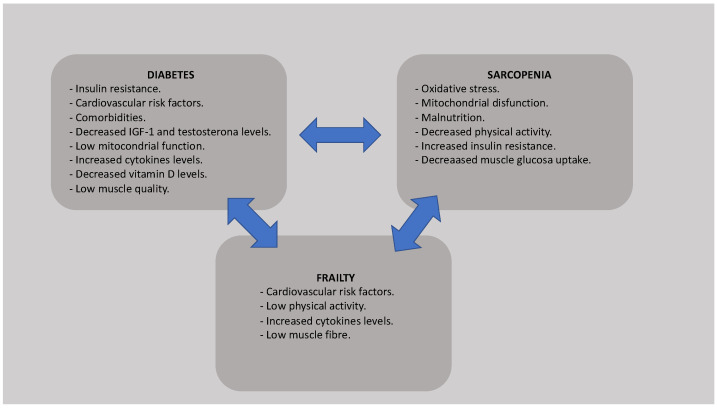
Pathophysiological link between diabetes, frailty, and sarcopenia.

**Table 1 ijerph-19-08677-t001:** Fried’s Frailty Criteria.

Findings
Involuntary weigh loss of 10 lbs or more in the last six months.
Reduced grip strength.
Difficulty initiation movements,
Reduced walking speed.
Fatigue.

**Frailty scale**: Fit (no abnormalities), Pre-frail (2 abnormalities or less), Frail (3 or more abnormalities).

**Table 2 ijerph-19-08677-t002:** Frequent causes of sarcopenia.

Frequent Causes of Sarcopenia
**Nutrition:** - Low protein intake - Low energy intake - Micronutrient deficiency - Malabsorption and other gastrointestinal conditions - Anorexia
**Associated: with inactivity:** - Bed rest, immobility deconditioning - Low activity, sedentary lifestyle
**Diseases:** - Cardiorespiratory disorders including chronic heart failure and chronic obstructive pulmonary disease - Metabolic disorders, particularly diabetes mellitus - Endocrine diseases, hormone deficiencies - Neurodegenerative diseases - Cancer - Liver and kidney disorders
**Others** - Hospital admission - Drug-relate - Ageing

**Table 3 ijerph-19-08677-t003:** Main non-pharmacological measures in T2DM with frailty and/or sarcopenia.

Main Non-Pharmacological Measures
Physical activity
Nutritional counseling
Improving mental health
Cognitive stimulation
Avoid hypoglycemia
Fostering social ties
Fall prevention

**Table 4 ijerph-19-08677-t004:** Main recommendation drugs in T2DM patients with frailty and/or sarcopenia.

Antidiabetic Drugs Recommended in Frail Patients	Antidiabetic Drugs Not Recommended in Frail Patients
Biguanides	Sulfonylureas
DDP-4 inhibitors	Meglintinides
Insulin (individualize)	Thiazolidinediones
	GLP-1 agonist
	SGLT-2 inhibitors

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
