# Peer review of "Management of Type 2 Diabetes Mellitus in Elderly Patients with Frailty and/or Sarcopenia"

_ijerph, 2022, doi:10.3390/ijerph19148677_

Round 1
Reviewer 1 Report
Type 2 Diabetes Mellitus (T2DM) is a chronic disease, and it also frequently coexists with sarcopenia and frailty in the elderly. The MS by Jaime Sanz-Cánovas and colleagues is focused on the management of T2DM in elderly patients with frailty/sarcopenia. Attempting to review this topic is not new. As there are a lot of reviews, papers, and books on the topic, it is difficult to present now a paper stimulating the interest of the reader in the field without new facts or new ideas.
Author Response
Dear Reviewer. We have tried to bring together in this review all the published evidence in order to facilitate the readers the management of these pathologies.
Reviewer 2 Report
Dear author
This review is important for managing DM patients with sarcopenia or frailty and providing present problems to study in the future. This review's present form deserves acceptable.
I would like to provide the below comments: 1) In page 4, physical activity section, I think that some of older patients with DM and sarcopenia or frailty sometimes cannot do aerobic exercise and resistance training recommended by the guideline. 2)Please introduce the method of exercise for those patients if there were articles studying the methods.
From reviewer.
Author Response
Dear reviewer, we are very pleased with your words and your assessment of our work. I attach a word document where I answer your questions. Thank you very much.

Reviewer 3 Report
Excellent review article on Management of type 2 diabetes mellitus patients awith frailty and sarcopenia. Good introduction of working hypothesis of Type 2 Diabetes in older patients as well as current definitions of sarcopenia and frailty. Explains pharmacological and non pharmacological treatments. Appropriate conclusion drawn.
Author Response
I am very grateful for your words and for the positive recognition of our work. Thank you very much.
Reviewer 4 Report
This review by Sanz-Cánovas et al. analyzed the potential role of diabetes in the development of frailty and sarcopenia, also focusing on pharmacological and not pharmacological strategies to prevent and improve sarcopenia and/or frailty in elderly patients with diabetes mellitus. Some issues to be addressed:
1) authors could more deeply explain why diabetes represents a significant risk factor for frailty and sarcopenia, reporting pathophysiological consequences of diabetes favouring frailty and sarcopenia development. The authors included some concepts on page 2, but they should go into detail to improve clarity for readers who are not experts in this field.
2) authors could clarify the pathophysiological link between diabetes, frailty and sarcopenia, using, for example, one or more figures explaining molecular pathways connecting these three pathological conditions.
3) in the paragraph "Glycemic Control Goals in the Elderly Patient", the authors examined the challenges of adequate glycaemic control in elderly patients, avoiding hypoglycaemic episodes that could increase mortality. They could probably add some practical suggestions from guidelines; moreover, they should underline the limitations related to standard glycaemic parameters such as fasting blood glucose and glycated haemoglobin, particularly in frail diabetic patients. In this specific setting, glycaemic variability has been proposed as a more comprehensive assessment of glycemic control.
4) authors could summarize non-pharmacological and pharmacological strategies to prevent and improve sarcopenia and frailty among diabetic patients with one or more figures.
5) Incretins have demonstrated anti-oxidative and anti-inflammatory properties, along with anti-thrombotic effects, that could be helpful in frailty patients (DOI:10.1186/s12933-018-0800-2; DOI: 10.3389/fphar.2021.670155). Furthermore, SGLT2 inhibitors, such as other new glucose-lowering agents, have demonstrated beneficial effects on renal outcomes (DOI: 10.1080/13543784.2020.1811231). Please add some comments on this.
6) Authors should improve the conclusions, including their work's potential limitations and future prospectives.
Author Response
Dear reviewer, I submit a word document with all my responses. I am very grateful about your recommendations.

Round 2
Reviewer 1 Report
Authors have now largely improved the manuscript.
Author Response
Dear reviewer.
Thank you very much for your words. We hope that our work is to your liking.
Best regards.
Reviewer 4 Report
The authors addressed all the reviewer’s comments, adding essential concepts to the main topic; however, two critical considerations:
- The authors should significantly improve the English language, and many typing errors are still in the manuscript.
- The authors might potentially improve the added figures and tables.
Author Response
Dear reviewer.
Thank you very much for your comments. Regarding the language, I have revised the manuscript and corrected some errors. I have made some modification in the tables and I have changed the figure.
We look forward to hearing from you. Thank you very much. Kind regards.